# On Consistent Bayesian Inference from Synthetic Data

## Abstract

Generating synthetic data, with or without differential privacy, has attracted significant attention as a potential solution to the dilemma between making data easily available, and the privacy of data subjects. Several works have shown that consistency of downstream analyses from synthetic data, including accurate uncertainty estimation, requires accounting for the synthetic data generation. There are very few methods of doing so, most of them for frequentist analysis. In this paper, we study how to perform consistent Bayesian inference from synthetic data. We prove that mixing posterior samples obtained separately from multiple large synthetic datasets converges to the posterior of the downstream analysis under standard regularity conditions when the analyst's model is compatible with the data provider's model. We show experimentally that this works in practice, unlocking consistent Bayesian inference from synthetic data while reusing existing downstream analysis methods.

## 1 Introduction

Synthetic data has the potential of opening privacy-sensitive datasets for widespread analysis. The idea is to train a generative model with real data, and release synthetic data that has been generated from the model. The synthetic data does not contain records from real people, and ideally it preserves the population-level properties of the real data, making it useful for analysis. Privacy preservation can be guaranteed with *differential privacy* (DP) (Dwork et al. 2006b), which offers provable protection of privacy.

The most convenient and straightforward way for downstream analysts to analyse synthetic data is using the same method that would be used with real data. However, ignoring the additional stochasticity arising from the synthetic data generation will yield biased results and overconfident uncertainty estimates (Raghunathan et al. 2003; Räisä et al. 2023; Wilde et al. 2021). This is especially problematic under DP, which requires adding extra noise, which will be ignored if the synthetic data is treated like real data. This problem creates the need for *noise-aware* analyses that account for the synthetic data generation.

When the downstream analysis is frequentist, it is possible to account for the synthetic data generation when multiple synthetic datasets are generated and analysed (Raghunathan et al. 2003). Recent work has extended this to DP synthetic data (Räisä et al. 2023), which allows generating multiple synthetic datasets without compromising on privacy. These methods reuse the analysis method for the real data, and only require using simple combining rules to combine the results from the analyses on each synthetic dataset, making them simple to apply.

For Bayesian downstream analyses, Wilde et al. (2021) have shown that the analyst can use additional samples of public real data to correct their analysis. However, their method requires targeting a generalised notion of the posterior (Bissiri et al. 2016) and needs the additional public data for calibration. Ghalebikesabi et al. (2022) propose a correction using importance sampling to avoid the

need of public data, but only prove convergence to a generalised posterior and do not clearly address the noise-awareness of the method.

In the context of missing data, Gelman et al. (2014) have proposed inferring the downstream posterior of a Bayesian analysis by imputing multiple completed datasets, inferring the analysis posterior for each completed dataset separately, and mixing the posteriors together. We study the applicability of this method to synthetic data, aiming the bring the simplicity of the frequentist methods using multiple synthetic datasets to Bayesian downstream analysis.

**Contributions**

1. We study inferring the downstream analysis posterior by generating multiple synthetic datasets, inferring the analysis posterior for each synthetic dataset as if it were the real dataset, and mixing the posteriors together. We find that in this setting, the synthetic datasets also need to be larger than the original dataset.

2. We prove that when the Bernstein–von Mises, or a similar theorem, applies, this method converges to the true posterior as the number of synthetic datasets and the size of the synthetic datasets grow. Under stronger assumptions, we prove a convergence rate for this method in the synthetic dataset size, which we expect to match the rate that usually applies in the Bernstein–von Mises theorem (Hipp and Michel 1976). These are presented in Section 3.

3. We evaluate this method with two examples in Section 4: non-private univariate Gaussian mean estimation, and differentially private Bayesian logistic regression. In the first example, we use the tractability of the model to derive further theoretical properties of the method, and in both examples, we verify that the method works in practice through experiments.

## 1.1 Related Work

Generating synthetic data to preserve privacy was, as far as we know, originally proposed by Liew et al. (1985). Rubin (1993) proposed accounting for the synthetic data generation in frequentist downstream analyses by adapting *multiple imputation* (Rubin 1987), which involves generating multiple synthetic datasets, analysing each of them, and combining the results with so called Rubin's rules (Raghunathan et al. 2003; Reiter 2002). Recently, Räisä et al. (2023) have shown that multiple imputation also works when the synthetic data is generated under DP when the data generation algorithm is *noise-aware* in a certain sense.

Wilde et al. (2021) study downstream Bayesian inference from DP synthetic data by considering the analyst's model to be misspecified, and targeting a generalised notion of the posterior (Bissiri et al. 2016) to deal with the misspecification, which makes method their more difficult to apply than standard Bayesian inference. They also assume that the analyst has additional public data available to calibrate their method.

Ghalebikesabi et al. (2022) use importance sampling to correct for bias with DP synthetic data, and have Bayesian inference as an example application. However, they also target a generalised variant (Bissiri et al. 2016) of the posterior instead of the noise-aware posterior we target, and they do not evaluate uncertainty estimation, so the noise-awareness of their method is not clear.

We are not aware of any existing work adapting multiple imputation for Bayesian downstream analysis in the synthetic data setting. In the missing data setting without DP, where multiple imputation was originally developed (Rubin 1987), Gelman et al. (2014) have proposed sampling the downstream posterior by mixing samples of the downstream posteriors from each of the multiple synthetic datasets. We find that this is not sufficient in the synthetic data setting, and add one extra component: our synthetic datasets are larger than the original dataset. We compare the two cases in more detail in Supplemental Section F, and in particular explain why large synthetic datasets are not needed in the missing data setting.

Noise-aware DP Bayesian inference is critical for taking into account the DP noise in synthetic data, but only a few works address this even without synthetic data. Bernstein and Sheldon (2018) present an inference method for simple exponential family models. Their approach was extended to linear models (Bernstein and Sheldon 2019) and generalised linear models (Kulkarni et al. 2021). Recently, Ju et al. (2022) developed an MCMC sampler that can sample the noise-aware posterior using a noisy summary statistic.

## 2 Background on Bayesian Inference

Bayesian inference is a paradigm of statistical inference where the data analyst's uncertainty in a quantity $Q$ after observing data $X$ is represented using the posterior distribution $p(Q|X)$ (Gelman et al. 2014). The posterior is given by Bayes' rule:

$$p(Q|X) = \frac{p(X|Q)p(Q)}{\int p(X|Q')p(Q')\,\mathrm{d}Q'}, \tag{1}$$

where $p(X|Q)$ is the likelihood of observing the data $X$ for a given value of $Q$, and $p(Q)$ is the analyst's prior of $Q$. Computing the denominator is typically intractable, so analysts often use numerical methods to sample $p(Q|X)$ (Gelman et al. 2014).

**Bernstein–von Mises Theorem**  It turns out that in many typical settings, the prior's influence on the posterior vanishes when the dataset $X$ is large. A basic example of this is the Bernstein–von Mises theorem (van der Vaart 1998), which informally states that under some regularity conditions, the posterior approaches a Gaussian that does not depend on the prior as the size of the dataset increases.

A crucial component of the theorem, and also our theory, is the notion of *total variation distance* between random variables, which is used to measure the difference between two random variables or probability distributions.

**Definition 2.1.** *The total variation distance between random variables (or distributions) $P_1$ and $P_2$ is*

$$\mathrm{TV}(P_1, P_2) = \sup_A |\Pr(P_1 \in A) - \Pr(P_2 \in A)|, \tag{2}$$

*where $A$ is any measurable set.*

As a slight abuse of notation, we allow the arguments of $\mathrm{TV}(\cdot, \cdot)$ to be random variables, probability distributions, or probability density functions interchangeably. We list some properties of total variation distance that we use in Lemma A.1 in the Supplement.

Now we can state the theorem.

**Theorem 2.2** (Bernstein–von Mises (van der Vaart 1998))**.** *Let $n$ denote the size of the dataset $X_n$. Under regularity conditions stated in Condition A.4 in Supplemental Section A.2, for true parameter value $Q_0$, the posterior $\bar{Q}(X_n) \sim p(Q|X_n)$ satisfies*

$$\mathrm{TV}\left(\sqrt{n}(\bar{Q}(X_n) - Q_0), \mathcal{N}(\mu(X_n), \Sigma)\right) \xrightarrow{P} 0 \tag{3}$$

*as $n \to \infty$ for some $\mu(X_n)$ and $\Sigma$, that do not depend on the prior, where the convergence in probability is over sampling $X_n \sim p(X_n|Q_0)$.*

## 3 Bayesian Inference from Synthetic Data

When the downstream analysis is Bayesian, and the analyst has access to non-DP synthetic data, they would ultimately want to obtain the posterior $p(Q|X, I_A)$ of some quantity $Q$ given real data $X$, where $I_A$ denotes the background knowledge such as priors of the analyst. In the DP case, the exact posterior is unobtainable, so we assume that $X$ is only available through a noisy summary $\tilde{s}$ (Ju et al. 2022; Räisä et al. 2023), so the posterior is $p(Q|\tilde{s}, I_A)$. To unify these notations, we use $Z$ to denote the observed values, so $Z = X$ in the non-DP case, $Z = \tilde{s}$ in the DP case, and the posterior of interest is $p(Q|Z, I_A)$. We summarise these random variables and their dependencies in Figure 1, and give an introduction to DP in Supplemental Section A.3.

In order to introduce the synthetic data into the posterior of interest, we can decompose the posterior as

$$p(Q|Z, I_A) = \int p(Q|Z, X^*, I_A)p(X^*|Z, I_A)\,\mathrm{d}X^*, \tag{4}$$

where we abuse notation by using $X^*$ as the variable to integrate over, so inside the integral $X^*$ is not a random variable. The decomposition in (4) means that we could sample $p(Q|Z, I_A)$ by first

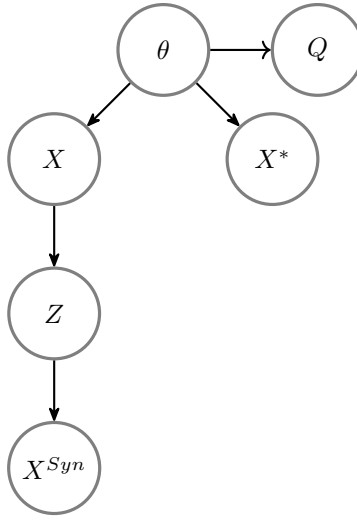

- $\theta$: data generating model parameters
- $X$: real data
- $X^*$: hypothetical data
- $Z$: observed summary of $X$ ($Z = X$ without DP)
- $X^{Syn}$: synthetic data, $X^{Syn} \sim p(X^*|Z, I_S)$
- $Q$: estimated quantity in downstream analysis
- $I_S$: synthetic data generator's background information
- $I_A$: analyst's background information

Figure 1: Left: random variables in noise-aware uncertainty estimation from synthetic data. Right: a Bayesian network describing the dependencies of the random variables.

sampling the synthetic data from the posterior predictive $X^{Syn} \sim p(X^*|Z, I_A)$, and then sampling $Q \sim p(Q|Z, X^* = X^{Syn}, I_A)$.

Note that the random variable $X^*$ represents a hypothetical real dataset that could be obtained if more data was collected, as seen in Figure 1, and it is not the synthetic dataset. The synthetic dataset $X^{Syn}$ is a sample from the conditional distribution of $X^*$ given $Z$. For this reason, $p(Q|Z, X^*, I_A) \neq p(Q|Z, I_A)$. To make our notation less cluttered, we write $p(\cdot|X^*, \cdot)$ in place of $p(\cdot|X^* = X^{Syn}, \cdot)$ in probabilities when the meaning is clear.

There are still two major issues with the decomposition in (4):

1. Sampling $p(Q|Z, X^*, I_A)$ requires access to $Z$, which defeats the purpose of using synthetic data.

2. $X^*$ needs to be sampled conditionally on the analyst's background information $I_A$, while the synthetic data provider could have different background information $I_S$.

To solve the first issue, in Section 3.2 we show that if we replace $p(Q|Z, X^*, I_A)$ inside the integral of (4) with $p(Q|X^*, I_A)$, the resulting distribution converges to the desired posterior,

$$\int p(Q|X^*, I_A) p(X^*|Z, I_A) \, \mathrm{d}X^* \to p(Q|Z, I_A) \tag{5}$$

in total variation distance as the size of each synthetic data set $X^*$ grows. It should be noted that many such synthetic data sets will be needed to account for the integral over $X^*$.

The second issue is known as *congeniality* in the multiple imputation literature (Meng 1994; Xie and Meng 2016). We look at congeniality in the context of Bayesian inference from synthetic data in Section 3.1, and find that we can obtain $p(Q|Z, I_A)$ under appropriate assumptions on the relationship between $I_A$ and $I_S$.

Exactly sampling the LHS of (5) requires generating a synthetic dataset for each sample of $p(Q|Z, I_A)$, which is not practical. However, we can perform a Monte-Carlo approximation for $p(Q|Z, I_A)$ by generating $m$ synthetic datasets $X_1^{Syn}, \ldots, X_m^{Syn} \sim p(X^*|Z, I_A)$, drawing multiple samples from each of the $p(Q|X^* = X_i^{Syn}, I_A)$, and mixing these samples, which allows us to obtain more than one sample of $p(Q|Z, I_A)$ per synthetic dataset. We look at some properties of this in Supplemental Section E, but we use the integral form in (5) in the rest of our theory.

## 3.1 Congeniality

In the decomposition (4) of the analyst's posterior, $X^*$ should be sampled conditionally on the analyst's background information $I_A$, while in reality the synthetic data provider could have different background information $I_S$.

A similar distinction has been studied in the context of missing data (Meng 1994; Xie and Meng 2016), where the imputer of missing data has a similar role as the synthetic data generator. Meng (1994) found that Rubin's rules implicitly assume that the probability models of both parties are compatible in a certain sense, which Meng (1994) defined as *congeniality*.

As our examples with Gaussian distributions in Section 4.1 and Supplemental Section C.2 show, some notion of congeniality is also required in our setting. However, because we study synthetic data instead of imputation, and Bayesian instead of frequentist downstream analysis, we need a different formal definition. As the analyst only makes inferences on $Q$, it suffices that both the analyst and synthetic data generator make the same inferences of $Q$:

**Definition 3.1.** *The background information sets $I_S$ and $I_A$ are congenial for observation $Z$ if*

$$p(Q|X^*, I_S) = p(Q|X^*, I_A) \tag{6}$$

*for all $X^*$ and*

$$p(Q|Z, I_S) = p(Q|Z, I_A). \tag{7}$$

In the non-DP case, (7) is redundant, as it is implied by (6), but in the DP case, both are needed, as the parties may draw different conclusions on $X$ given $Z = \tilde{s}$.

Combining congeniality and (5),

$$\int p(Q|X^*, I_A)p(X^*|Z, I_S)\, \mathrm{d}X^* = \int p(Q|X^*, I_S)p(X^*|Z, I_S)\, \mathrm{d}X^*$$
$$\to p(Q|Z, I_S) = p(Q|Z, I_A), \tag{8}$$

where the convergence is in total variation distance as the size of $X^*$ grows. In the following, we assume congeniality, and drop $I_A$ and $I_S$ from our notation.

## 3.2 Consistency Proof

To recap, we want to prove that the posterior from synthetic data,

$$\bar{p}_n(Q) = \int p(Q|X_n^*)p(X_n^*|Z)\, \mathrm{d}X_n^*, \tag{9}$$

converges in total variation distance to $p(Q|Z)$ as the size $n$ of $X_n^*$ grows. We prove this in Theorem 3.4, which requires that both $p(Q|Z, X_n^*)$ and $p(Q|X_n^*)$ approach the same distribution as $n$ grows. We formally state this in Condition 3.2. In Lemma 3.3, we show that Condition 3.2 is a consequence of the Bernstein–von Mises theorem (Theorem 2.2) under some additional assumptions, so we expect it to hold in typical settings.

To make the notation more compact, let $\bar{Q}_n^+ \sim p(Q|Z, X_n^*)$, and let $\bar{Q}_n \sim p(Q|X_n^*)$.

**Condition 3.2.** *For all $Q$ there exist distributions $D_n$ such that*

$$\mathrm{TV}\left(\bar{Q}_n^+, D_n\right) \xrightarrow{P} 0 \quad \text{and} \quad \mathrm{TV}\left(\bar{Q}_n, D_n\right) \xrightarrow{P} 0 \tag{10}$$

*as $n \to \infty$, where the convergence in probability is over sampling $X_n^* \sim p(X_n^*|Z, Q)$.*

Theorem 2.2 implies Condition 3.2 with some additional assumptions:

**Lemma 3.3.** *If the assumptions of Theorem 2.2 (Condition A.4) and the following assumptions:*

*(1) $Z$ and $X^*$ are conditionally independent given $Q$; and*

*(2) $p(Z|Q) > 0$ for all $Q$,*

*hold for the downstream analysis for all $Q_0$, then Condition 3.2 holds.*

*Proof.* The full proof is in Supplemental Section B.1. Proof idea: when $Z$ and $X^*$ are conditionally independent given $Q$,

$$p(Q|Z, X^*) \propto p(X^*|Q)p(Z|Q)p(Q) \tag{11}$$

so $p(Q|Z, X^*)$ can be equivalently seen as the result of Bayesian inference with observed data $X^*$ and prior $p(Q|Z)$. As the only difference to $p(Q|X^*)$ is the prior, the Bernstein–von Mises theorem implies that both $p(Q|Z, X^*)$ and $p(Q|X^*)$ converge in total variation distance to the same distribution. $\qquad\square$

Assumption (1) of Lemma 3.3 will hold if the downstream analysis treats its input data as an i.i.d. sample from some distribution. Assumption (2) holds when the likelihood is always positive, and in the DP case when the density of the privacy mechanism is positive everywhere, which is the case for common DP mechanisms like the Gaussian and Laplace mechanisms (Dwork and Roth 2014).

Next is the main theorem of this work: (5) holds under Condition 3.2.

**Theorem 3.4.** *Under congeniality and Condition 3.2,* $\mathrm{TV}\left(p(Q|Z), \bar{p}_n(Q)\right) \to 0$ *as* $n \to \infty$.

*Proof.* The full proof is in Supplemental Section B.1. Proof idea: the proof consists of three steps. The first two are in Lemma B.1 and the third is in Lemma B.2 in the Supplement. The first step is showing that $\mathrm{TV}(\bar{Q}_n, \bar{Q}_n^+) \xrightarrow{P} 0$ when $X_n^* \sim p(X_n^*|Z, Q)$ for fixed $Z$ and $Q$. This is a simple consequence of the triangle inequality and Condition 3.2, as total variation distance is a metric. In the second step, we show that $\mathrm{TV}(\bar{Q}_n, \bar{Q}_n^+) \xrightarrow{P} 0$ also holds when $X_n^* \sim p(X_n^*|Z)$. In the final step, we show that this implies the claim. $\qquad\square$

### 3.3 Convergence Rate

Under stronger regularity conditions, we can get a convergence rate for Theorem 3.4. The regularity conditions depend on uniform integrability:

**Definition 3.5.** *A sequence of random variables $X_n$ is uniformly integrable if*

$$\lim_{M \to \infty} \sup_n \mathbb{E}(|X_n|\mathbb{I}_{|X_n|>M}) = 0 \tag{12}$$

Now we can state the regularity conditions for a convergence rate $O(R_n)$:

**Condition 3.6.** *There exist distributions $D_n$ such that for a sequence $R_1, R_2, \cdots > 0$, $R_n \to 0$ as $n \to \infty$,*

$$\frac{1}{R_n} \mathrm{TV}\left(\bar{Q}_n^+, D_n\right) \quad \text{and} \quad \frac{1}{R_n} \mathrm{TV}\left(\bar{Q}_n, D_n\right) \tag{13}$$

*are uniformly integrable when $X_n^* \sim p(X_n^*|Z)$.*

Note that $X_n^* \sim p(X_n^*|Z)$ conditions on $Z$, not $Q$ and $Z$ like in Condition 3.2. We prove that Condition 3.6 is met in univariate Gaussian mean estimation for $R_n = \frac{1}{\sqrt{n}}$ in Theorem D.1 in the Supplement. This is the same rate that commonly applies in the Bernstein–von Mises theorem (Hipp and Michel 1976).

Condition 3.6 implies an $O(R_n)$ convergence rate:

**Theorem 3.7.** *Under congeniality and Condition 3.6,* $\mathrm{TV}\left(p(Q|Z), \bar{p}_n(Q)\right) = O(R_n)$.

*Proof.* The full proof is in Supplemental Section B.2. Proof idea: first, we prove the uniform integrability of $\frac{1}{R_n} \mathrm{TV}(\bar{Q}_n, \bar{Q}_n^+)$ when $X_n^* \sim p(X_n^*|Z)$ by using the triangle inequality and properties of uniform integrability. Second, we prove that this implies the claimed convergence rate. $\qquad\square$

## 4 Examples

In this section, we present two examples of downstream inference from synthetic data at a high level. First, we demonstrate univariate Gaussian mean estimation. Second, we have logistic regression on a toy dataset, with DP synthetic data. In the first example, we use the tractability of the model to derive additional theoretical properties, and in both examples, we experimentally verify our theory.

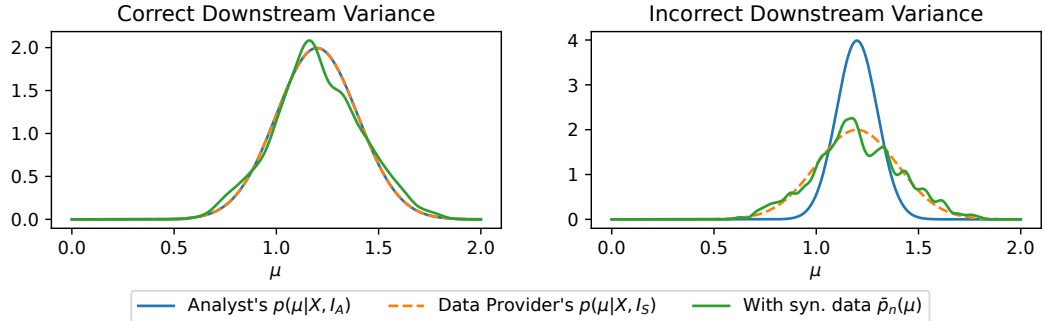

Figure 2: Simulation results for the Gaussian mean estimation example, showing that the mixture of posteriors from synthetic data in green converges. In the left panel, both the analyst and data provider have the correct known variance. The blue and orange lines overlap, as both parties have the same $p(\mu|X)$. On the right, the analyst's known variance is too small ($\hat{\sigma}_k^2 = \frac{1}{4}\bar{\sigma}_k^2$), so congeniality is not met, but the mixture of posteriors from synthetic data, $\bar{p}_n(\mu)$, still converges to the data provider's posterior. In both panels, $m = 400$ and $\frac{n_{X*}}{n_X} = 20$.

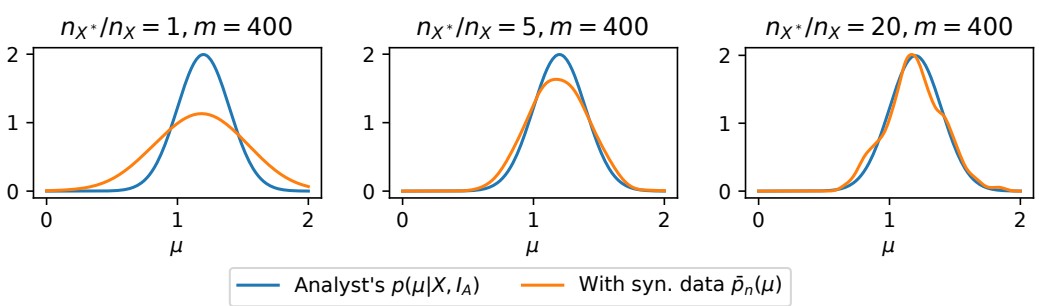

Figure 3: Convergence of the mixture of posteriors from synthetic data with different sizes of the synthetic dataset on Gaussian mean estimation with known variance. $n_{X*} = n_X$ is clearly not enough, but $n_{X*} = 20n_X$ is already relatively good.

Supplemental Section C, contains more detailed descriptions of the examples, and some additional results. Supplemental Section D proves an $O(\frac{1}{\sqrt{n}})$ convergence rate for Theorem 3.4 in the Gaussian mean estimation case. Our code is in the supplementary material.

### 4.1 Non-private Gaussian Mean Estimation

Our first example is very simple: the analyst infers the mean $\mu$ of a Gaussian distribution with known variance from synthetic data that has been generated from the same model. The posteriors for this setting can be found in Supplemental Section A.4. To differentiate the variables for the analyst and data provider, we use bars for the data provider (like $\bar{\sigma}_0^2$) and hats for the analyst (like $\hat{\sigma}_0^2$).

When the synthetic data is generated from the known variance model with known variance $\bar{\sigma}_k^2$, we sample from the posterior predictive $p(X^*|X)$ as

$$\bar{\mu}|X \sim \mathcal{N}(\bar{\mu}_{n_X}, \bar{\sigma}_{n_X}^2), \quad X^*|\bar{\mu} \sim \mathcal{N}^{n_{X*}}(\bar{\mu}, \bar{\sigma}_k^2) \tag{14}$$

$$\bar{\mu}_{n_X} = \frac{\frac{1}{\bar{\sigma}_0^2}\bar{\mu}_0 + \frac{n_X}{\bar{\sigma}_k^2}\bar{X}}{\frac{1}{\bar{\sigma}_0^2} + \frac{n_X}{\bar{\sigma}_k^2}}, \quad \frac{1}{\bar{\sigma}_{n_X}^2} = \frac{1}{\bar{\sigma}_0^2} + \frac{n_X}{\bar{\sigma}_k^2}. \tag{15}$$

$\mathcal{N}^{n_{X*}}$ denotes a Gaussian distribution over $n_{X*}$ i.i.d. samples.

When downstream analysis is the model with known variance $\hat{\sigma}_k^2$, we have

$$\hat{\mu}|X^* \sim \mathcal{N}(\hat{\mu}_{n_{X^*}}, \hat{\sigma}_{n_{X^*}}^2), \quad \hat{\mu}_{n_{X^*}} = \frac{\frac{1}{\hat{\sigma}_0^2}\hat{\mu}_0 + \frac{n_{X^*}}{\hat{\sigma}_k^2}\bar{X}^*}{\frac{1}{\hat{\sigma}_0^2} + \frac{n_{X^*}}{\hat{\sigma}_k^2}}, \quad \frac{1}{\hat{\sigma}_{n_{X^*}}^2} = \frac{1}{\hat{\sigma}_0^2} + \frac{n_{X^*}}{\hat{\sigma}_k^2}. \tag{16}$$

Now, using $\mu^*$ to denote a sample from the mixture of posteriors from synthetic data $\bar{p}_n(\mu)$ in (9), we show in Supplemental Section C.1 that

$$\mathbb{E}(\mu^*) \to \bar{\mu}_{n_X}, \quad \text{Var}(\mu^*) \to \bar{\sigma}_{n_X}^2 \tag{17}$$

as $n_{X^*} \to \infty$, so $\mu^*$ asymptotically has the same mean and variance as the downstream posterior distribution $p(\mu|X)$ on the real data.

We test the theory with a numerical simulation in Figure 2. We generated the real data $X$ of size $n_X = 100$ by i.i.d. sampling from $\mathcal{N}(1, 4)$. Both the analyst and data provider use $\mathcal{N}(0, 10^2)$ as the prior. The data provider uses the correct known variance ($\bar{\sigma}_k^2 = 4$), and the analyst either uses the correct known variance ($\hat{\sigma}_k^2 = 4$), or a too small known variance ($\hat{\sigma}_k^2 = 1$), which is an example of uncongeniality.

In the congenial case in the left panel of Figure 2, both parties have the same posterior given the real data $X$, and the mixture of posteriors from synthetic data is very close to that. In the uncongenial case in the right panel, where the analyst underestimates the variance, the parties have different posteriors given $X$, but the mixture of synthetic data posteriors is still close to the data provider's posterior.

In Figure 3, we examine the convergence of the mixture of posteriors from synthetic data under congeniality. We see that setting $n_{X^*} = n_X$ is not enough, as the mixture of posteriors is significantly wider than the analyst's posterior. The synthetic dataset needs to be larger than the original, with $n_{X^*} = 5n_X$ already giving a decent approximation and $n_{X^*} = 20n_X$ a rather good one. In Figure S1 in the Supplement, we also examine the effect of $m$ on the mixture of synthetic data posteriors, and see that $m$ must also be sufficiently large, otherwise the method produces very jagged posteriors.

## 4.2  Differentially Private Logistic Regression

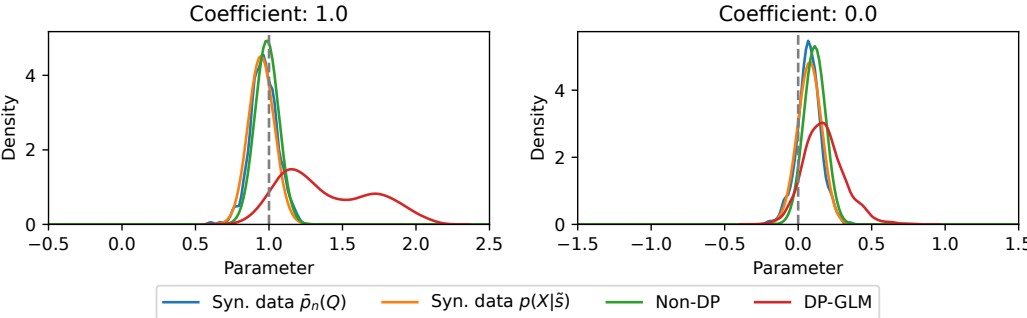

Figure 4: Posteriors in the DP logistic regression experiment, where $Q$ are the regression coefficients. The mixture of posteriors from synthetic data, $\bar{p}_n(Q)$, (with $n_{X^*}/n_X = 20$, $m = 400$) is very close the to the private posterior $p(Q|\tilde{s})$ computed using (4). Computing the posterior without synthetic data with DP-GLM gives a somewhat wider posterior. The true parameter values are highlighted by the grey dashed lines and shown in the panel titles. The privacy bounds are $\epsilon = 1$, $\delta = n_X^{-2} = 2.5 \cdot 10^{-7}$.

Our second example is logistic regression on a simple 3-d binary toy dataset, ($n_X = 2000$), with DP synthetic data, under the same setting as used by Räisä et al. (2023) for frequentist logistic regression. We change the downstream task to Bayesian logistic regression to evaluate our theory.

Under DP, $Z$ is a noisy summary $\tilde{s}$ of the real data. We need synthetic data sampled from the posterior predictive $p(X^*|\tilde{s})$, which is exactly what the NAPSU-MQ algorithm of Räisä et al. (2023) provides. In NAPSU-MQ, $\tilde{s}$ is the values of user-selected marginal queries with added Gaussian noise. We used the open-source implementation of NAPSU-MQ[1] by Räisä et al. (2023), and describe NAPSU-MQ in Supplemental Section A.3.

---

[1] https://github.com/DPBayes/NAPSU-MQ-experiments

Because of the simplicity of this model, it is possible to use the exact posterior decomposition (4) as a baseline, by using $p(X|\tilde{s})$ instead of $p(X^*|\tilde{s})$ to generate synthetic data. We give a detailed description of this process in Supplemental Section C.5. We have also included the DP-GLM algorithm (Kulkarni et al. 2021) that does not use synthetic data, and the non-DP posterior from the real data as baselines. We obtained the code for DP-GLM from Kulkarni et al. (2021) upon request.

Figure 4 compares the mixture of posteriors from synthetic data from (9) that uses $p(Q|X^*)$, with $n_{X^*}/n_X = 20$ and $m = 400$ synthetic datasets, to the baselines. The posterior from (9) is very close to the posterior from (4). The DP-GLM posterior that does not use synthetic data is somewhat wider. The privacy bounds are $\epsilon = 1, \delta = n_X^{-2} = 2.5 \cdot 10^{-7}$.

We ran the experiment 100 times and also with $\epsilon = 0.1$ and $\epsilon = 0.5$, and plot coverages and widths of credible intervals in Figure S4 in the Supplement. With $\epsilon = 1$ and $\epsilon = 0.5$, the coverages are accurate and DP-GLM consistently produces wider intervals. With $\epsilon = 0.1$, the mixture of synthetic data posteriors likely needs more and larger synthetic datasets to converge, as it produced wider and slightly overconfident intervals for one coefficient.

## 5   Discussion

Synthetic data are often considered as a substitute for real data that are sensitive. Since the data generation process is based on having access to the $Z$, one might ask why is the synthetic data needed in first place. Why cannot we simply perform the downstream posterior analysis directly using $Z$? Our analysis allows $Z$ to be an arbitrary, even noisy, representation of the data, and it might be difficult for the analyst to place a model for such generative process for $Q$. In most applications, the analyst does have a model for $Q$ arising from the data. Therefore using the synthetic data as a proxy for the $Z$ allows the analyst to use existing models and inference methods to perform the analysis.

**Limitations**   A clear limitation of mixing posteriors from multiple synthetic datasets is the computational cost of analysing many large synthetic datasets, which may be substantial for more complex Bayesian downstream models, where even a single analysis can be computationally expensive. However, the separate analyses can be run in parallel. We also expect the downstream posteriors of different synthetic datasets to be similar to each other, so it should be possible to use the information gained from sampling a few of them to speed up sampling the others.

Under DP, we need noise-aware synthetic data generation, which limits the settings in which the method can currently be applied. However, if new noise-aware methods are developed in the future, the method can immediately be used with them.

To recover the analyst's posterior, the method requires congeniality, which basically requires the analyst's prior to be compatible with the data provider's. However, the method was still able to recover the data provider's posterior in the Gaussian example, suggesting that the data provider's prior information overrides the analyst's prior information. This suggests an interesting area of future research: analysis methods that override the data provider's prior. An importance sampling approach similar to that of Ghalebikesabi et al. (2022) could provide one approach. This observation also raises interesting questions on whether truly general and objective synthetic data generation is possible.

**Conclusion**   We considered the problem of consistent Bayesian inference of downstream analyses using multiple, potentially DP, synthetic datasets, and studied an inference method that mixes the posteriors from multiple large synthetic datasets. We proved, under general and well-understood regularity conditions of the Bernstein–von Mises theorem, that the method is asymptotically exact as the sizes of the synthetic datasets grow. We also derived a convergence rate under stricter regularity conditions. We studied the method in two examples: non-private Gaussian mean estimation and DP logistic regression. In the former, we were able to use the analytically tractable structure of the setting to derive additional properties of the method, including a convergence rate without additional assumptions. In both settings, we experimentally validated our theory, and showed that the method works in practice. This fills a major gap in the synthetic data analysis literature, unlocking consistent Bayesian inference while reusing existing downstream analysis methods.

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
