# OpenReview forum: "On Consistent Bayesian Inference from Synthetic Data"
_NeurIPS.cc/2023/Conference — Submitted to NeurIPS 2023_

### Official Review · Reviewer_ZbGV · 2023-07-05

**Soundness:** 1 poor
**Presentation:** 1 poor
**Contribution:** 3 good
**Rating:** 3
**Confidence:** 2

**Summary:**

The authors consider the use of synthetic data $X^{sync} $ created from a model $p(X^{sync}|Z, I_S)$ on a further Bayesian analysis where the analyst has access to $p(Q|X^{sync})$ and to $p(X^{sync}| Z)$, where $Q$ is the parameter.
 the paper is based on equality (4)
$ p(Q|Z) = \int p(Q|Z.X^*)p(X^*|Z)dx^*$ where $X^*$ essentially represents $X^{sync}$ and $Z$ is either the real data or a differentially private version of the data and the idea is to replace
$\int p(Q|Z.X^*)p(X^*|Z)dx^*$ by $p_n(Q) = \int p(Q|X^*)p(X^*|Z)dx^*$ which is fully accessible by the Bayesian and then show that the latter is close to the former.

To do that the authors assume that
$p(Q|Z,X^*_n)$ and $p(Q|X^*_n)$ are both close in total variation to the same seq of distribution say $D_n$ in probability when $X^*_n $ follows $p(X^*_n|Z,Q_0)$ and that given $Q$ $X^*$ and $Z$ are independent.

Then the authors treat 2 toy examples a Gaussian and a logistic regression example and run some simulations to illustrate.

Inthe above presentation I am not mentioning the fact that the Bayesian models can be different from the generating model, which is treated but quickly pushed by assuming that this is not a problem.

**Strengths:**

The paper is well motivated and it is an important problem. If the results are correct then the paper is relevant and interesting.

**Weaknesses:**

I am not sure the results are correct. From the presentation I don't understand the author's eq (4) or rather their comment which says that
$p(Q|X^*, Z) $ is different from $p(Q|Z)$. The reason is that the authors do not explain what is the generating model for $X^*$ (In their paper the authors sometimes use $X^{sync}$ and sometimes $X^*$ as if they were the same, so I gather that they represent the same thing , but in the DAG of fig 1, they are not the same at all.

The $X^* = X^{sync}$ is distributed from $p(X^*|Z)$, which does not depend on $Q$. Hence unless the authors clarify this point then their result are not valid.

The authors consider two toy examples but in neither of them do they check tht the theoretical setup considered before is valid. For instance in the Gaussian example the generating model for the synthetic data is
$ X^* \sim p( \cdot | X) = \int p(\cdot | \mu) \pi(\mu| X) d\mu$, i.e. the posterior predictive density.
The relation (4) writes as $\pi(\mu|X) = \int p(\mu | X, x^*)p(x^*|X)dx^*$ but the model $p(\mu | X, x^*)$ is not defined. The authors seem to consider that  $x^* , X|\mu$ are iid but this is not possible because $\mu$ is unknown and it does not correspond to their Gaussian example.

There are a number of other results which seem dubious to me. See below.

**Questions:**

1. In condition 3.2 the authors write for all Q but do not mention $Z$ while the distribution depends on $Z$. Is the condition almost sure in Z? in probability ? The same is true for condition 3.6.

2. Lemma 3.3 : What does it mean " hold for the downstream analysis for all $Q_0$ " ? condition (1) says given $Q$. Are $Q$ and $Q_)$ the same?
3. eq (17) says that they have the same mean and variance but not that they have the same limiting distribution. Why don't  the authors verify the assumptions for this toy example. Surely if these assumptions don't hold for that one they will never hold.

4. in the supplement equation (204) : in my version of Asymptotic statistics there is not corollary 2.3 but a continuous mapping Theorem. I imagine that the authors are referring to the Lehman - Scheffe Theorem which states that if the sequence of probability densities $f_n$ converges pointwise to a probability density $f$ then it converges in $L_1$. Hoewever here the sequence is also random and the convergence is pointwise almost surely and I don't see the argument which allows to glue all these sets (for each Q) of probabiulity 1 to apply Lehman Scheffe. In other words the sets of proba 1 may differ from one $Q$ to another.

5. Minor comments: The authors recall in the call fairly trivial probability results which they should quote and recall possibly in the supplement and free this space to better explain their setup.

**Limitations:**

The authors are conscious of some of the limitations of their method.

---

> ### Author Rebuttal · Authors · 2023-08-09
>
> > I am not sure the results are correct. From the presentation I don't understand the author's eq (4) or rather their comment which says
> that $p(Q | X^*, Z)$
> is different from $p(Q | Z)$. The reason is that the authors do not explain what is the generating model for $X^*$(In their paper the authors sometimes use $X^{sync}$ and sometimes $X^*$
> as if they were the same, so I gather that they represent the same thing , but in the DAG of fig 1, they are not the same at all.
>
> > The $X^* = X^{sync}$ is distributed from $p(X^* | Z)$, which does not depend on $Q$. Hence unless the authors clarify this point then their result are not valid.
>
> $X^*$ and $X^{Syn}$ are not the same thing, as we note on line 133. $X^*$ is a random variable representing a
> hypothetical dataset that could
> be obtained if more data was collected, so its generating model given $\theta$ is the same as for the real data $X$.
> The synthetic data $X^{Syn}$ is a sample from the posterior predictive distribution
> $p(X^* | Z) = \int p(X^* | \theta)p(\theta | Z)d \theta$.
>
> This distinction between $X^*$ in the posterior predictive
> distribution and a sample of the posterior predictive is not limited to synthetic data: it is
> present in all Bayesian analyses when the posterior predictive is involved. The distinction leads to the Bayesian
> network in Figure 1, where $X^*$ is not a child of $Z$, but $X^{Syn}$ is. The interpretation of $X^*$ as a hypothetical
> real dataset makes it clear why $p(Q | X^*, Z) \neq p(Q | Z)$, as the former is conditioning on more information than
> the latter. We will add additional clarification of this to Section 3, and will clarify some parts
> where $X^*$ was referred to as a synthetic dataset.
>
> > The authors consider two toy examples but in neither of them do they check tht the theoretical setup considered before is valid. For instance in the Gaussian example the generating model for the synthetic data is $X^* \sim p(\cdot | X) = \int p(\cdot | \mu)\pi(\mu | X) d\mu$, i.e. the posterior predictive density. The relation (4) writes as
> $\pi(\mu | X) = \int p(\mu | X, x^*)p(x^*|X)d x^*$
> but the model $p(\mu | X, x^*)$ is not defined. The authors seem to consider that $x^*, X | \mu$ are iid but this is not possible because $\mu$ is unknown and it does not correspond to their Gaussian example.
>
> Because $X^*$ represents a hypothetical real dataset, $p(\mu | X, X^*)$ is simply the posterior when concatenating
> $X$ and $X^*$ together into a single dataset. While $X^{Syn}$ and $X$ are obviously not independent given $\mu$,
> $X^*$ and $X$ are independent given $\mu$. We don't see why $\mu$ being unknown would affect these independencies.
>
> > 1. In condition 3.2 the authors write for all Q but do not mention $Z$ while the distribution depends on $Z$. Is the condition almost sure in Z? in probability ? The same is true for condition 3.6.
>
> In both cases, the condition is for the specific value of $Z$ which is observed. We will add this to the condition
> statements.
>
> > 2. Lemma 3.3 : What does it mean " hold for the downstream analysis for all $Q_0$
> " ? condition (1) says given $Q$. Are $Q$ and $Q_)$ the same?
>
> $Q_0$ refers to Theorem 2.2 and Condition A.4. We will reorder the words of the statement to make it less confusing.
>
> > 3. eq (17) says that they have the same mean and variance but not that they have the same limiting distribution. Why don't the authors verify the assumptions for this toy example. Surely if these assumptions don't hold for that one they will never hold.
>
> We can verify the assumptions of Lemma 3.3 for this example. The only assumptions that are not obvious are
> (2-4) in Condition A.4. As the Gaussian likelihood has a score function, it is differentiable in quadratic mean
> (van der Vaart 1998), so (2) holds. The Fisher information is straightforward to calculate from
> its definition, and is $\frac{1}{\hat{\sigma}^2_k} \neq 0$ in this case, so (3) holds.
> For (4), we can set $\phi_n$ to reject (output 1) when $|\frac{\bar{X}_n - \mu_0}{\sigma}| \geq \frac{1}{2}\beta$
> and accept (output 0) otherwise. We also note that there is ample literature on the assumptions of the Bernstein-von Mises
> theorem, and our additional assumptions in Lemma 3.3 are much easier to check, which we discuss on lines 197-200.
>
> We can also see that $\mu^*$ has a Gaussian distribution, as the other distributions involved are Gaussian, which we will add to the paper.
>
> > 4. in the supplement equation (204) : in my version of Asymptotic statistics there is not corollary 2.3 but a continuous mapping Theorem. I imagine that the authors are referring to the Lehman - Scheffe Theorem [...]. Hoewever here the sequence is also random and the convergence is pointwise almost surely and I don't see the argument which allows to glue all these sets (for each Q) of probabiulity 1 to apply Lehman Scheffe. [...]
>
> We are indeed referring to a special case of the Lehman-Scheffe theorem, which appears as corollary
> 2.30 in (van der Vaart 1998). It states that if $X_n$ and $X$ are random vectors with densities
> $p_n$ and $p$ with respect to a measure $\mu$, and if $p_n \to p$ pointwise $\mu$-almost everywhere, then $X_n$ converges to
> $X$ in total variation, meaning that $\lim_{n\to \infty} \mathrm{TV}(X_n, X) = 0$. Our argument then is the
> following: for any fixed $X^*_{i,n}$ such that (201) holds, (204) will hold due to the aforementioned corollary, as
> the sequence of densities is not random with fixed $X^*_{i,n}$. When $X^*_{i,n} \sim p(X^*_n|Z)$, (201) holds almost
> surely, so the previous argument yields (204) almost surely, which is what we claim.
>
> >5.  Minor comments: The authors recall in the call fairly trivial probability results which they should quote and recall possibly in the supplement and free this space to better explain their setup.
>
> If this is referring to Section 2, we think the material there is important for the rest of the paper, especially for readers who are not experts on Bayesian inference or probability theory.

---

> ### Comment · Area_Chair_C9Rd · 2023-08-17
> **what did you think of the authors' response?**
>
> The authors responded to a few of your questions/concerns. Can you take a look and say whether your perspective on the submission has changed?
>
> In particular, the authors say that $X^*$ and $X^{sync}$ are not equal. Did their response/clarification on this change your mind?

---

### Official Review · Reviewer_8977 · 2023-07-07

**Soundness:** 2 fair
**Presentation:** 3 good
**Contribution:** 2 fair
**Rating:** 3
**Confidence:** 3

**Summary:**

The paper works on performing consistent Bayesian inference from synthetic data under DP. The authors propose a solution that involves mixing posterior samples from multiple large synthetic datasets, proving that this technique converges to the posterior of downstream analysis under specific conditions. This was established through experimentation involving non-private Gaussian mean estimation and DP logistic regression.

**Strengths:**

The paper offers a unique and engaging exploration of Bayesian Inference in the context of Synthetic Data, providing a fresh perspective in a field predominantly characterized by frequentist analysis.

**Weaknesses:**

See questions.

**Questions:**

The motivation behind the research is not explicitly stated. Could you clarify the unique benefits that Bayesian Inference offers in this context? How does it enhance the study or application beyond the capabilities of other methodologies (frequentist)?

The rationale for using Synthetic Data, specifically Synthetic Data without DP, is also vague. This choice seems to offer no additional solid protection under this setting. When applying DP, why choose to release Synthetic Data instead of the DP summary directly?

The paper's primary theoretical contribution is not evidently defined. The claim that the distribution of Synthetic Data can be arbitrarily close to the original distribution as the sample size 'n' approaches infinity appears to be a trivial expectation. Could you elaborate on this aspect more?

There has been prior discussion on the topic of Bayesian Inference from Synthetic Data, for example, as seen in reference [1]. Could you specify what new insights or advancements your study brings to the table, beyond the contributions of these previous works?



[1] Wilde, H., Jewson, J., Vollmer, S., & Holmes, C. (2021, March). Foundations of Bayesian learning from synthetic data. In International Conference on Artificial Intelligence and Statistics (pp. 541-549). PMLR.


**Limitations:**

While the study explores the concept of Synthetic Data, its impetus is not distinctly articulated, leading to ambiguity regarding the problem the authors aim to address. The theoretical contribution appears to be weak.

---

> ### Author Rebuttal · Authors · 2023-08-09
>
> > The motivation behind the research is not explicitly stated. Could you clarify the unique benefits that Bayesian Inference offers in this context? How does it enhance the study or application beyond the capabilities of other methodologies (frequentist)?
>
> Bayesian inference is widely used in numerous statistical applications because of the flexibility it allows for statistical modelling (multilevel models, partial pooling, incorporating prior knowledge).
> A popular Bayesian regression modelling package brms, for example, has been cited more than 5500 times (Google Scholar) since its publication in 2017, with more than 1150 citations in 2023 by early August.
> In theory, our results could be used to easily translate any of these applications and many more to use (private) synthetic data.
>
> > The rationale for using Synthetic Data, specifically Synthetic Data without DP, is also vague. This choice seems to offer no additional solid protection under this setting. When applying DP, why choose to release Synthetic Data instead of the DP summary directly?
>
> We fully agree that DP is required for synthetic data to provide strong privacy protection.
> We included the non-DP case because our theory covers both cases in almost the same way, and this could be of independent theoretical interest. Releasing synthetic data instead of the summary makes the job of the analyst much easier, as they can directly reuse their existing analysis methods and code by just using synthetic data instead of real data. Using the summary directly would require to analyst to develop a new model based on observing the summary,
> which could take significant effort.
>
> > The paper's primary theoretical contribution is not evidently defined. The claim that the distribution of Synthetic Data can be arbitrarily close to the original distribution as the sample size 'n' approaches infinity appears to be a trivial expectation. Could you elaborate on this aspect more?
>
> Our primary theoretical contribution is studying if multiple synthetic datasets could be used for
> consistent downstream Bayesian inference, finding that they can by mixing the downstream analysis posterior samples from
> multiple large synthetic datasets, and proving that distribution obtained from this converges to
> the desired distribution under our assumptions. The size of the real dataset is fixed in our
> theory, only the synthetic dataset's size grows to infinity.
>
> While the result may seem trivial in retrospect, the requirement of increasing synthetic data sizes was a surprise to us as that is not required in the frequentist setting. Furthermore, the formal proof of convergence is certainly a non-trivial undertaking.
>
> > There has been prior discussion on the topic of Bayesian Inference from Synthetic Data, for example, as seen in reference [1]. Could you specify what new insights or advancements your study brings to the table, beyond the contributions of these previous works?
>
> As we mention in the Related Work-section, the paper by Wilde et al. (2021) calibrates a posterior from synthetic data
> using public data, which would lead to a significantly weaker privacy model than our fully DP model.
> They also target a generalised variant of the posterior, which could make applying their method
> harder. Our method targets the standard notion of posterior, and uses the multiple large synthetic datasets for calibration
> instead, so public data is not needed.

---

> > ### Comment · Reviewer_8977 · 2023-08-18
> >
> > Thank you for your response. I still have a few concerns:
> >
> > 1: The authors assert that "Releasing synthetic data instead of the summary ... reuse their existing analysis methods and code".. However, in the context of mechanisms based on data perturbation for Differential Privacy, conventional methods often require specialized treatment, such as debiasing or correction. The notion of directly applying existing analysis methods to synthetic data while disregarding the underlying data raises concerns. This approach may seem straightforward, but it could potentially lead to problems. Take, for instance, the task of computing the sample median under DP. Well-established techniques like those outlined in [1] have been developed. Nevertheless, as far as my understanding goes, achieving similar accuracy under DP with synthetic data insertion poses challenges. For the sample median task, the idea of "reusing the existing analysis method" might suggest generating a synthetic dataset and applying the sample median as the "existing analysis method." However, the results may not align meaningfully with those achieved in [1]. The examples presented by the authors rely on strong assumptions (such as Gaussian models). Moreover, in Gaussian settings, utilizing noised parameters instead of synthetic data could be more convenient. (Generating synthetic data from parameter is easy. Sending the parameter is easier than sending synthetic data).
> >
> > 2: If I grasp the concept accurately, the authors contend that both the sizes of synthetic data and the number of synthetic datasets need to approach infinity, a requirement not present in the frequentist setting. This condition appears to be a notable disadvantage. Is there an underlying lower bound that demonstrates the inevitability? Additionally, why should we opt for the Bayesian setting if this challenge doesn't manifest in the frequentist context?
> >
> >
> > [1] Smith, A. (2011, June). Privacy-preserving statistical estimation with optimal convergence rates. In Proceedings of the forty-third annual ACM symposium on Theory of computing (pp. 813-822).

---

> > > ### Author Response · Authors · 2023-08-19
> > >
> > > Thank you for taking the time to read our rebuttal.
> > >
> > > > However, in the context of mechanisms based on data perturbation for Differential Privacy, conventional methods often require specialized treatment, such as debiasing or correction. The notion of directly applying existing analysis methods to synthetic data while disregarding the underlying data raises concerns.
> > >
> > > The point of our theory is to provide this kind of correction to many types of downstream analyses at the same time, while making the correction easy to use, as the non-DP  analysis method can be reused.
> > >
> > >
> > > > Take, for instance, the task of computing the sample median under DP. Well-established techniques like those outlined in [1] have been developed. Nevertheless, as far as my understanding goes, achieving similar accuracy under DP with synthetic data insertion poses challenges. For the sample median task, the idea of "reusing the existing analysis method" might suggest generating a synthetic dataset and applying the sample median as the "existing analysis method." However, the results may not align meaningfully with those achieved in [1].
> > >
> > > We agree that using a tailored DP method should beat using DP synthetic data
> > > and doing the downstream analysis on the synthetic data, for any single
> > > analysis task. However, once the allocated privacy budget has been exhausted
> > > with tailored DP analyses, the original data has to be thrown away, never to
> > > be used again, which severely limits the practical applicability of such
> > > methods. Synthetic data allows an arbitrary number of analyses to be
> > > done on the synthetic data due to the post-processing immunity of DP.
> > >
> > > It also cannot be taken for granted that a tailored DP method
> > > always beats synthetic data + non-DP method. In our UCI Adult experiment,
> > > mixing the synthetic data posteriors clearly beat the DPVI which doesn't
> > > use synthetic data, and the alternative method DP-GLM was not able to
> > > fully converge.
> > >
> > > > The examples presented by the authors rely on strong assumptions (such as Gaussian models). Moreover, in Gaussian settings, utilizing noised parameters instead of synthetic data could be more convenient. (Generating synthetic data from parameter is easy. Sending the parameter is easier than sending synthetic data).
> > >
> > > The Gaussian example is only meant to serve as the simplest possible example,
> > > where the analytical tractability of the setting allows checking
> > > variour properties of the mixture of synthetic datasets analytically,
> > > such as the effect of uncongeniality. Our theory applies to much more complex
> > > settings, where the downstream method may not have sufficient statistics
> > > that could be published instead of the synthetic data. In addition, just
> > > publishing sufficient statistics would limit the analyses that can be done,
> > > while synthetic data allows arbitrary analyses.
> > >
> > > > 2: If I grasp the concept accurately, the authors contend that both the sizes of synthetic data and the number of synthetic datasets need to approach infinity, a requirement not present in the frequentist setting. This condition appears to be a notable disadvantage. Is there an underlying lower bound that demonstrates the inevitability?
> > >
> > > We are not aware of any such lower bound, but it is clear from all our experiments that having synthetic data sets of equal size as the real data leads to overestimating posterior variances.
> > > We were able to derive an
> > > additional variance correction for the Gaussian mean estimation which
> > > allows approximating the effect of large synthetic datasets in Supplemental
> > > Section C.4, so it is possible to get around the requirement in at least
> > > that case.
> > >
> > > > Additionally, why should we opt for the Bayesian setting if this challenge doesn't manifest in the frequentist context?
> > >
> > > Bayesian inference has the advantages over frequentist inference
> > > we mentioned in our
> > > rebuttal (multilevel models, partial pooling, incorporating prior knowledge).
> > > As Bayesian inference is such a widely used paradigm, providing methods to
> > > use it with synthetic data is useful and allowing practitioners to see
> > > what the tradeoffs are is useful, even if Bayesian inference doesn't end
> > > up being the right choice for every setting.
> > >
> > > Ultimately the aim of our paper is to increase theoretical understanding of what is and what is not possible for Bayesian inference with synthetic data. This understanding will help researchers make better choices between different approaches and develop new even better ones. We believe that results showing what is not possible can be extremely useful in this sense.

---

> ### Comment · Area_Chair_C9Rd · 2023-08-17
> **what did you think of the authors' response?**
>
> The authors have provided responses to your questions and comments. Please revise the text and score of your review to reflect how their responses have changed your perspective on their submission, and please acknowledge that you have read the authors' carefully written response.

---

### Official Review · Reviewer_4qww · 2023-07-07

**Soundness:** 3 good
**Presentation:** 2 fair
**Contribution:** 3 good
**Rating:** 6
**Confidence:** 2

**Summary:**

Inspired by Bayesian approaches for performing multiple imputation of missing data, this paper investigates the applicability of similar strategies for the analysis of synthetic data. Namely, the paper proposes inferring the downstream posterior of a Bayesian analysis by: generating multiple synthetic datasets; inferring the analysis posterior for each synthetic dataset; and mixing the posteriors together. (Interestingly, the paper finds that, contrary to the missing data imputation context, in the synthetic data case this strategy requires the synthetic datasets to be larger than the original dataset.)

The paper provides theory showing that under the regularity conditions of the Bernstein-von Mises theorem (augmented by the additional conditions presented in Lemma 3.3), and assuming the congenial conditions in Definition 3.1, then the proposed strategy will approximate the data provider posterior distribution as the number of synthetic datasets and the synthetic dataset sizes increase. (The paper also proves a convergence rate result under stronger assumptions.)

The method is evaluated using two simple examples: (i) non-private univariate Gaussian mean estimation (when the variance is assumed to be known); and (ii) differentially private logistic regression.


**Strengths:**

This is an interesting paper. It addresses an important topic with an approach that appears to be novel and sound.

**Weaknesses:**

One limitation of the proposed approach appears to be its reliance on the congeniality assumption (which we should not expect to hold in general). While the paper uses a simple example to illustrate that the method was still able to recover the data provider’s posterior when congeniality was violated, the paper needs to provide more extensive evidence of the robustness of the proposed approach w.r.t. violations of this assumption (as, in practice, it seems that the usefulness of the proposed approach for data analysis will depend on how robust the method is to violations of congeniality).

More specifically, the paper shows that for the toy problem of Gaussian mean estimation (with known variance) the mixture of posteriors converges to the data provider’s posterior even when the analyst’s variance is different from the data provider’s variance (right panel of Figure 2). However, for this example we have that the posterior distribution for the mean is already Gaussian in the finite sample setting to begin with. Providing additional examples where the posterior distribution of the quantity of interest is not Gaussian in the finite sample setting, but where the mixture of posteriors approximates the data provider’s posterior when congeniality is violated would provide more convincing illustrative examples. Perhaps, one simple example is the problem of Gaussian variance (or precision) estimation with known means. In this case, the paper could assess the robustness w.r.t. congeniality violations by choosing different means for the data provider and data analyst.  The paper should provide additional examples along these lines.

The paper might also want to include some discussion about some practically important settings where the Bernstein-von Mises theorem does not hold, and where the proposed approach might not be applicable (e.g., for models where the number of parameters increases with the sample size).

Other minor suggestions:

Line 69: change “which makes method their more” to “which makes their method more”

Line 302: change “To recover the analyst’s posterior …” to “To recover the data provider’s posterior …”


**Questions:**

See suggestions above.

**Limitations:**

Yes, the paper addresses well the limitations of the proposed method.

---

> ### Author Rebuttal · Authors · 2023-08-09
>
> > One limitation of the proposed approach appears to be its reliance on the congeniality assumption (which we should not expect to hold in general). While the paper uses a simple example to illustrate that the method was still able to recover the data provider’s posterior when congeniality was violated, the paper needs to provide more extensive evidence of the robustness of the proposed approach w.r.t. violations of this assumption (as, in practice, it seems that the usefulness of the proposed approach for data analysis will depend on how robust the method is to violations of congeniality).
>
> > More specifically, the paper shows that for the toy problem of Gaussian mean estimation (with known variance) the mixture of posteriors converges to the data provider’s posterior even when the analyst’s variance is different from the data provider’s variance (right panel of Figure 2). However, for this example we have that the posterior distribution for the mean is already Gaussian in the finite sample setting to begin with. Providing additional examples where the posterior distribution of the quantity of interest is not Gaussian in the finite sample setting, but where the mixture of posteriors approximates the data provider’s posterior when congeniality is violated would provide more convincing illustrative examples. Perhaps, one simple example is the problem of Gaussian variance (or precision) estimation with known means. In this case, the paper could assess the robustness w.r.t. congeniality violations by choosing different means for the data provider and data analyst. The paper should provide additional examples along these lines.
>
> We worked through the suggested example of Gaussian variance estimation with known mean.
> Specifically, the model assumes that the data $X = (x_1, \dotsc, x_n)$ is generated from a
> Gaussian distribution with some known mean $\mu_k$, and the task is to estimate the variance
> $\sigma^2$ of the Gaussian. In this example, the synthetic data provider knows the correct
> mean, but the analyst may use an incorrect known mean.
>
> It turns out that the mixture of synthetic data posteriors does not recover the data provider's posterior in this case.
> Specifically, the mixture's mean in the limit of infinite synthetic data is larger than
> the data provider's mean by the square of the
> difference between the known means of the parties. Interestingly, the shape and variance of the
> of the mixture empirically appear to converge to the data provider's posterior, so only the
> means are different, as seen in Figure R4 of the attached file. However, we haven't had time to verify this mathematically.
>
> While this example shows that the mixture of synthetic data posteriors is not always robust to
> congeniality violations, the the experiment with Adult data presented in the general response provides
> evidence that congeniality violations may not be an issue in practice, as the mixture of
> synthetic data posteriors was still much closer to the real data non-DP posterior than the baseline
> DPVI.
>
> In any case, our main contribution is the theory studying how consistent Bayesian inference
> could be done from multiple synthetic datasets, which includes finding that congeniality even is
> an issue in this setting, and treating the congenial case. The uncongenial case is an important
> direction of future work, and we think that our paper can serve as a good starting point for it.
>
> > The paper might also want to include some discussion about some practically important settings where the Bernstein-von Mises theorem does not hold, and where the proposed approach might not be applicable (e.g., for models where the number of parameters increases with the sample size).
>
> We will add some examples to the limitations section where the Bernstein-von Mises theorem
> may not hold, including models with increasing numbers of parameters, infinite-dimensional
> models, and models with support that heavily depends on the parameters.

---

> > ### Comment · Reviewer_4qww · 2023-08-19
> >
> > Thank you for your thoughtful and detailed responses to all my questions (i.e., robustness to violations of the congeniality assumption and examples where the Bernstein-von Mises theorem does not hold).
> >
> > I appreciate your response regarding violations of the congeniality assumption for the Gaussian variance example. The failure of the method on this simple example decreases its practical appeal. Also, while the real data illustration provides some evidence that these violations might not be an issue in practice, this is again just one example, and more extensive illustrations would be needed to assess this point. (So, I suggest the authors include a fairly nuanced discussion about the robustness of their method w.r.t. violations of this assumption in the final version of the paper.)
> >
> > That being said, I also feel the authors make a fair point when they argue that the paper’s main contribution is the theory. In this regard, I agree the paper already provides enough contributions for a first publication and represents a first step towards more complicated settings that can be addressed in future work.
> >
> > Overall, I am still leaning towards the acceptance of the paper (but might change my mind depending on the outcome of these final discussions with other reviewers).

---

> > > ### Author Response · Authors · 2023-08-21
> > >
> > > Thank you for taking the time to read our rebuttal. We are planning to include the Gaussian variance example in the revised version, with extended discussion on congeniality.

---

> ### Comment · Area_Chair_C9Rd · 2023-08-17
> **what did you think of the authors' response?**
>
> The authors have provided detailed responses to your questions and comments. Please revise the text and score of your review to reflect how their responses have changed your perspective on their submission, and please acknowledge that you have read the authors' carefully written response.

---

### Official Review · Reviewer_uV6S · 2023-07-08

**Soundness:** 3 good
**Presentation:** 3 good
**Contribution:** 3 good
**Rating:** 5
**Confidence:** 2

**Summary:**

The paper studies Bayesian inference based on synthetic datasets generated in a DP and non-DP setting.

The paper suggests a specific sampling approach for downstream Bayesian inference using synthetic DP and non-DP dataset. It contributes theoretical results on the convergence of the inferenced posterior (from synthetic dataset) to the true posterior showing that (under certain assumptions) it converges as the the of number of synthetic datasets and the size of the datasets increases. Additionally, a convergence rate is derived
Experimental results are provided for non-DP Bayesian mean inference and DP Bayesian logistic regression showing that the inference approach works, along with examples of the effect of parameters influencing the convergence (including number of observations, samples and level of congeniality)


**Strengths:**

- Very timely and interesting topic; I enjoyed learning about the specific Bayesian+DP setting.
- The theory is mostly well presented in the main paper (see suggestion/questions below). The narrative is relatively easy to follow.
- The theory appears sound. I have not found any obvious issues; however I would need to rely on other reviewers (and perhaps later the community as a whole) to validate the many proofs in the supplementary.


**Weaknesses:**

The following are question and comments; not necessarily weaknesses per se:

  -  The balance between theory and experiments is generally very good for my taste, but I feel the experimental part (in the main paper) let the theory part down a bit. The initial experiments focus on intuition and basic insights which I string support; however once the basics have been presented it would had been helpful with an experiment which covers many more scenarios proving summaries of the performance (using TV, coverage, means/modes and variance as metrics), along the most relevant dimensions such as number of observations, level of
  -  ... it would also have been interesting with a more realistic example (high dimensional) using a more complicated model to better motivate the paper. Have the authors validated the results on such an example?
  -  Figure 4 (right): It is not clear to me why the non-DP posterior does not mange to center its mode closer to the true parameter, is this an effect of the prior being centered on the true parameter combined with relatively few observations (the prior is not specified in the main text as far as I can tell?) - or other things?
  -  Figure 1: I am slightly confused by the graphical model, probably because the nature and role of $\theta$ is never really explained in detail. I hope the authors can clarify this (perhaps along with a detailed explanation the generative model in general)? For completeness, I would suggest including $I_a$ and $I_s$ in the figure as well.

Overall, I am generally positive about the paper but the experimental parts misses an opportunity to convince me. I will opt for a borderline score until I get a chance to see the other reviews and the authors' response.



**Questions:**

Included above.

**Limitations:**

Included above.

---

> ### Author Rebuttal · Authors · 2023-08-09
>
> > The balance between theory and experiments is generally very good for my taste, but I feel the experimental part (in the main paper) let the theory part down a bit. The initial experiments focus on intuition and basic insights which I string support; however once the basics have been presented it would had been helpful with an experiment which covers many more scenarios proving summaries of the performance (using TV, coverage, means/modes and variance as metrics), along the most relevant dimensions such as number of observations, level of
>
> We have added a plot of the total variation distance between the mixture of synthetic data
> posteriors and the target posterior for different numbers and sizes of synthetic datasets in the
> toy data experiment. See the general response for more details. We will add more of the suggested experiments in the final version if space allows.
>
> > ... it would also have been interesting with a more realistic example (high dimensional) using a more complicated model to better motivate the paper. Have the authors validated the results on such an example?
>
> We have added an experiment on the UCI Adult dataset. See the general response for details.
>
> > Figure 4 (right): It is not clear to me why the non-DP posterior does not mange to center its mode closer to the true parameter, is this an effect of the prior being centered on the true parameter combined with relatively few observations (the prior is not specified in the main text as far as I can tell?) - or other things?
>
> The downstream prior for the logistic regression example was indeed missing from the paper. It's a centered Gaussian with
> standard deviation $\sqrt{10}$ with two independent components. We've added a mention of it to Supplemental Section C.5.
> The distance of the non-DP posterior from the true parameters in the toy data experiment, shown
> in Figure 4,
> is simply due to randomness in sampling the relatively small
> number of datapoints. Frequentist logistic regression (implemented by Statsmodels) gives almost identical coefficients
> to the mean of the non-DP posterior when run with the same input dataset.
>
> > Figure 1: I am slightly confused by the graphical model, probably because the nature and role of $\theta$
>     is never really explained in detail. I hope the authors can clarify this (perhaps along with a detailed explanation the generative model in general)? For completeness, I would suggest including $I_a$ and $I_s$
>     in the figure as well.
>
> $\theta$ is the parameter(s) of the data generating model used by the data provider, so it appears in the
> posterior predictive $p(X^* | Z) = \int p(X^* | \theta)p(\theta | Z) d X^*$ that the data provider uses to sample
> the synthetic data. We will clarify this in the paper. $I_S$ and $I_A$ affect most of the nodes
> in the network, so adding the required edges would clutter the network. We will add a note that
> the whole network is conditioned on either one of them to the caption.

---

> > ### Comment · Reviewer_uV6S · 2023-08-18
> >
> > Thanks to the authors for addressing my questions (with an additional experiment and metrics) and what seems like a detailed rebuttal to other reviewers.
> > I am struggling with my assessment of this paper, but I still find it interesting and have not seen substantiated arguments against accepting it, so I’ll keep my score for now.
> >
> > For now, I note that the authors have made commitments to update the paper in several places; I feel it would be helpful with a concise list/summary of changes so we (the reviewers) can get an overview of the (key) updates suggested.
> >
> > Surprising results are the most interesting, and it would probably be helpful to the discussion if the authors could elaborate on the comment to reviewer 8977  “…the requirement of increasing synthetic data sizes was a surprise to us as that is not required in the frequentist setting”.

---

> > > ### Author Response · Authors · 2023-08-19
> > >
> > > Thank you for taking the time to read our rebuttal.
> > > We have made a summary of the paper updates as a general comment at the
> > > top.
> > >
> > > We found the requirement of increasing synthetic dataset sizes surprising,
> > > as it is not required in the frequentist setting studied by
> > > Räisä et al. (2023), or the Bayesian inference from missing data setting
> > > detailed in Supplemental Section F of our paper, which served as an
> > > inspiration to our work. When we started working on the problem, we did
> > > not see any reason why the synthetic dataset size should have such a
> > > different effect from the other two settings. Even now after
> > > writing the paper, the only indications of this difference we see are
> > > writing out Equation (4) and considering when $p(Q | Z, X^*, I_A)$ can
> > > be replaced with $p(Q, X^*, I_A)$, and the experiments which confirm this empirically.

---

> ### Comment · Area_Chair_C9Rd · 2023-08-17
> **what did you think of the authors' response?**
>
> The authors have provided detailed responses to your questions and comments. Please revise the text and score of your review to reflect how their responses have changed your perspective on their submission, and please acknowledge that you have read the authors' carefully written response.

---

### Official Review · Reviewer_z4aQ · 2023-07-28

**Soundness:** 2 fair
**Presentation:** 3 good
**Contribution:** 2 fair
**Rating:** 5
**Confidence:** 1

**Summary:**

This work is sloving a interesting task, which infers the downstream analysis posterior using synthetic data. The work proved that the Bernstein-von Mises theroy applies, the method can converage to the ture posterio as the number of synthetic datasets. The experimental settings are under two examples, i.e. non-private univariate Gaussian
56 mean estimation and differentially private Bayesian logistic regression.

**Strengths:**

1, The work is trying to solve an interesting task, which is infering the downstream analysis posterior using synthetic data.

2, The paper is well-writen and presented.

3, The code is provided. So it will be helpful for the following work.

**Weaknesses:**

1. Since synthetic data is generated by models which are trained using real data. So why synthetic data can improve the consisten bayesian inference is not clear. I think the paper needs more discussion about differences bewteen the real data and synthetic data.

2, The synthetic data is a big topic. In the work, for me, it is not clear which synthetic data methods are used and how the synthetic data method is trained using real data.

3, The applications are missing. Is it possible to extend the proposed method or therory to some kind of real application.

**Questions:**

See weaknesses.

**Limitations:**

See weaknesses.

---

> ### Author Rebuttal · Authors · 2023-08-09
>
> > Since synthetic data is generated by models which are trained using real data. So why synthetic data can improve the consisten bayesian inference is not clear. I think the paper needs more discussion about differences bewteen the real data and synthetic data.
>
> The synthetic data does not improve the downstream Bayesian inference over using the real data, so the real data
> should be used if it is available. The paper examines what can be done when the real data is not available due to
> privacy concerns, but synthetic data is available. We will clarify this in the paper.
> Proper privacy protection requires DP synthetic data which is our main focus, but we have included the non-private synthetic data setting as the same theory applies there as well.
>
> > The synthetic data is a big topic. In the work, for me, it is not clear which synthetic data methods are used and how the synthetic data method is trained using real data.
>
> Our main contribution is the theoretical result, which works for any synthetic data generation method capable of generating from the posterior predictive distribution, which is targeted for example by van Breugel et al. (ICML 2023).
> In the examples presented in the paper, the synthetic data in the Gaussian example is generated from the posterior predictive, as detailed in equations (14)
> and (15). In the logistic regression example, we used the NAPSU-MQ algorithm (Räisä et al., AISTATS 2023), which we've briefly described in
> Supplemental Section A.3.
>
> > The applications are missing. Is it possible to extend the proposed method or therory to some kind of real application.
>
> We have added an experiment on the UCI Adult dataset. See the general response for details.
> Our theory is not specific to any single synthetic data generation method, so it is
> potentially applicable to any release of privacy-sensitive synthetic data, such as medical data.

---

> > ### Comment · Reviewer_z4aQ · 2023-08-21
> >
> > After checking the responses, my concerns have been addressed. So I will to increase my score.

---

### Author Rebuttal · Authors · 2023-08-09

## Motivation and Contribution
Several reviewers wrote that the paper's contribution and motivation is unclear. Our motivation was
investigating whether multiple synthetic datasets could be used for consistent downstream Bayesian
inference when the real data is not available due to privacy concerns. Our main contributions
are finding that this is possible by mixing the posteriors from multiple large synthetic datasets,
and rigorously proving that this converges to the desired posterior under our assumptions.

We note the Bayesian viewpoint to synthetic data use has very recently received more
attention in the context of downstream prediction tasks. van Breugel et al. (2023) independently
proposed
aggregating downstream predictions from multiple synthetic datasets, and empirically observed
that this improves generalisation performance and uncertainty quantification. We will add discussion
on this to Related Work.

## Additional Experiments
Several reviewers asked for more experiments, especially on real data. We have conducted an experiment
on the UCI Adult dataset, in the same setting Räisä et al. (2023) used to evaluate NAPSU-MQ, which is the algorithm we used to generate synthetic data in this experiment, and the toy data experiment.
Specifically, synthetic data is generated under DP from a subset of the columns in the real data,
with the downstream task being logistic regression on a futher subset of the columns in the synthetic
data.

In our experiment, the logistic regression is Bayesian, and the posteriors from multiple synthetic
datasets are mixed together. The ideal target posterior is intractable in this setting, so we compare
against a non-DP posterior from the real data, and the DP variational inference (DPVI) algorithm of
Jälkö et al. (2017). We also tried DP-GLM, which was used in the toy data experiment, but were not
able to get useful posteriors out of it.

We have included a subset of the results in the attached file. In Figure R1, we plot the
posteriors from one run of the experiment with $\epsilon=1$. The mixture of synthetic data
posteriors $\bar{p}_n(Q)$ is fairly close to the real data non-DP posterior, with the exception
of two coefficients that correspond to the races with the smallest number of people in the data.
This is caused by the fact that NAPSU-MQ adds the same amount of noise to all categories, so
the signal-to-noise ratio is smaller for underrepresented groups. The posteriors from DPVI are
much less accurate. In Figure R2, we plot credible interval coverages from 20 repeats with
$\epsilon=1$. The mean of the non-DP Laplace approximation is considered the true value for
the coverage. $\bar{p}_n(Q)$ has much better coverage than DPVI. The small mismatch of coverage of
$\bar{p}_n(Q)$ is likely due to the fact that running NAPSU-MQ with all possible queries
would be computationally intractable in this setting, so the synthetic data has to lose some
information.

One reviewer asked for additional metrics, so we plotted the total variation distances between
the mixture of synthetic data posteriors and the target posterior for different numbers and
sizes of synthetic datasets in the toy data logistic regression experiment. These are shown
in Figure R3 for $\epsilon = 1$. We computed the total variation distances separately for
the 1D marginals, as computing the required integral over the 2D joint distribution took too long.
The results in the top row panels show that as the size of the synthetic dataset increases, the total variation distance
initially decreases at a steady rate, but stops decreasing at some point due to the finite number of
synthetic datasets. As the number of synthetic datasets increases, this plateau moves further,
and swapping the roles of the number and size of the synthetic datasets shows that adding more
synthetic datasets also decreases the total variation distance, which is seen on the bottom
row panels. We also plotted these with
$\epsilon = 0.5$ and $\epsilon = 0.1$, and will include them in the paper.

### References
- B. van Breugel, Z. Qian and M. van der Schaar. "Synthetic Data, Real Errors: How (Not) to Publish and Use Synthetic Data" ICML 2023
- J. Jälkö, O. Dikmen and A. Honkela. "Differentially Private Variational Inference for Non-conjugate Models" UAI 2017

---

### Comment · Area_Chair_C9Rd · 2023-08-17
**acknowledge and discuss the authors' detailed response**

The authors have provided a **very detailed** response, which includes answers to reviewers' questions and more exposition of the technical material. **Drop a comment in this thread with how their response changed your perspective on the submission.**

Remember to also update your scores/reviews to reflect how your views have changed in light of the authors' response, and respond to comments/questions the authors have left underneath individual reviews.

---

### Author Response · Authors · 2023-08-19
**Summary of Changes**

Summary of the changes we will make to the final version, requested by reviewer uV6S:

- Additional experiments
    - UCI Adult data (see general rebuttal)
    - Plots with total variation distance (see general rebuttal)
    - Example of Gaussian variance estimation with known mean (see rebuttal to reviewer 4qww)

- Discuss recent related work the Bayesian viewpoint to synthetic data (see general rebuttal)
- Clarify why synthetic data is useful (see rebuttal to reviewer z4aQ)
- Mention prior for the downstream logistic regression on toy data (see rebuttal to reviewer uV6S)
- Clarify role of $\theta$ (see rebuttal to reviewer uV6S)
- Clarify $I_A$ and $I_S$ in Figure 1 (see rebuttal to reviewer uV6S)
- Add example settings where the Bernstein-von Mises theorem does not hold to Limitations (see rebuttal to reviewer 4qww)
- Clarify the difference between $X^{Syn}$ and $X^*$ (see rebuttal to reviewer ZbGV)
- Clarify statements of Condition 3.2, Condition 3.6 and Lemma 3.3 (see rebuttal to reviewer ZbGV)
- Show that $\mu^*$ has a Gaussian distribution in the Gaussian mean estimation example (see rebuttal to reviewer ZbGV)

---

### Decision · Program_Chairs · 2023-09-21

**Decision:**

Reject

**Comment:**

Reviewers questioned the correctness of the paper, in particular its appeal to the Lehman-Scheffe Theorem, which the discussion session was not enough to properly sort out. Other reviewers were not entirely satisfied with the empirical evaluations, or confused by the paper's motivations. The authors did respond to these points, but there was not enough time to clear everything up sufficiently. I recommend the authors spend time editing to their paper to clarify these points and going through another round of submissions.